# A Peptide HEPFYGNEGALR from *Apostichopus japonicus* Alleviates Acute Alcoholic Liver Injury by Enhancing Antioxidant Response in Male C57BL/6J Mice

**DOI:** 10.3390/molecules27185839

**Published:** 2022-09-08

**Authors:** Qiliang Zhu, Huiling Zhuo, Lamei Yang, Haohong Ouyang, Jun Chen, Bing Liu, Hongliang Huang

**Affiliations:** 1School of Pharmacy, Guangdong Pharmaceutical University, Guangzhou 510006, China; 2School of Nursing, Guangdong Pharmaceutical University, Guangzhou 510006, China; 3School of Biosciences & Biopharmaceutics, Guangdong Pharmaceutical University, Guangzhou 510006, China; 4Key Specialty of Clinical Pharmacy, The First Affiliated Hospital of Guangdong Pharmaceutical University, Guangzhou 510006, China; 5Key Laboratory of New Drug Discovery and Evaluation, Guangdong Pharmaceutical University, Guangzhou 510006, China; 6Guangzhou Key Laboratory of Construction and Application of New Drug Screening Model Systems, Guangdong Pharmaceutical University, Guangzhou 510006, China

**Keywords:** sea cucumber, oxidative stress, autophagy, acute alcoholic liver injury

## Abstract

Liver-related disease caused by alcohol is a frequent disorder of the hepatic tract. Heavy consumption of alcohol in a short period causes oxidative damage to the liver. Sea cucumber is abundant in nutrients and its various extracts have been studied for antioxidant properties. One peptide was isolated and identified from *Apostichopus japonicus* in our recent study. We investigated the benefits of the peptide in a model of acute ethanol-induced male C57BL/6J mice. Dietary intake of the peptide could attenuate hepatomegaly, hepatitis and the accumulation of lipid droplets, and increase antioxidant enzyme activities in mice with acute alcoholic liver injury. The results indicated that a 20 mg/kg peptide supplement could activate the Nrf2/HO-1 pathway and block the nuclear translocation of NF-κB to alleviate oxidative stress and inflammation. In addition, the preventive effects of peptide supplementation may be related to autophagy. This study suggests that dietary supplementation with a sea cucumber-derived peptide is one of the potential candidates to alleviate acute alcoholic liver injury.

## 1. Introduction

Alcohol is one of the most popular beverages worldwide, but heavy drinking can lead to severe organ disease. One of the organs most sensitive to alcohol is the liver. Large volumes of alcohol intake can lead to acute alcoholic liver injury with severe pathological changes in the liver [1].

Reactive oxygen species (ROS) play a crucial role in acute alcoholic liver injury [2]. ROS can be produced in mitochondria, endoplasmic reticulum, nucleus and plasma membrane [3]. Excessive ROS can be triggered by a large intake of alcohol, causing oxidative stress [4]. Excessive ROS generated in hepatocytes will take more electrons from macromolecules such as DNA, protein, and lipids, resulting in oxidative damage [5]. ROS and lipid peroxides can also stimulate Kupffer cells to secrete various cytokines (such as tumor necrosis factor-α, interleukin and interferon) to aggravate inflammation in the liver, even leading to hepatic injury or death [6]. Some researchers believe that it is feasible to decrease the overproduction of ROS to prevent and treat alcohol-related liver disease [7,8].

Autophagy is a process for the degradation of proteins in which the intracellular cytoplasmic components are enveloped into autophagosomes for delivery to lysosomes or vacuoles for degradation [9]. Autophagy is classified as selective or non-selective. Non-selective autophagy typically occurs during the starvation of cells, which results in the breaking down of cytoplasmic components that supply the nutrients for cells. Selective autophagy targets misfolded protein aggregates, damaged and excess organelles, and lipid droplets. Impaired mitophagy (selective autophagy for damaged/excess mitochondria) has been known to be associated with alcoholic liver disease [10]. Blocking selective autophagy leaves cells more susceptible to oxidative damage [11]. This has the potential to prevent acute alcoholic liver injury, either with antioxidants for repairing autophagic defects, or by enhancing the autophagic capacity of the hepatocytes.

Antioxidants have been reported to alleviate oxidative stress and inflammation in alcoholic liver-related diseases. Now different drugs such as metadoxine, N-acetylcysteine and reduced glutathione can be utilized to enhance the antioxidant capacity of hepatocytes and accelerate liver regeneration. However, there are some side effects and drug resistance to the long-term dose. Peripheral neuritis has occurred in a few patients with long-term metadoxine. Cough, bronchial spasms and gastritis can be induced in some patients with N-acetylcysteine. Safer antioxidants are needed. The antioxidant properties of foodstuffs and the purified bioactive compounds used in supplement formulations have been extensively investigated. For instance, *rosmarinic acid*, an extract from *Rosmarinus officinalis*, has been reported to ameliorate H_2_O_2_-induced oxidative stress on L02 cells through the MAPK and Nrf2 pathways [12]. Sea cucumber is known for its abundant nutrients. In China, sea cucumbers have been used as dietary supplements for more than 2000 years [13], and their health benefits are partly attributed to their repertoire of proteins. Several studies have also reported that animal-derived peptides such as tuna and Alaskan pollock exert antioxidant effects in different studies [14,15]. Some investigations describe that the peptides extracted from sea cucumbers have antioxidants [16], ACE-reducing activity [17], anti-tumor [18] and antibacterial [19] effects, and other biological activities. Our recent study isolated three peptides identified from *Apostichopus japonicus* (*A. japonicus*) and revealed that they could alleviate oxidative stress in neuroblastoma cells [20]. Of the three peptides mentioned above, we chose one peptide to investigate its beneficial effects. The sequence of the chosen peptide was His-Glu-Pro-Phe-Tyr-Gly-Asn-Glu-Gly-Ala-Leu-Arg (HEPFYGNEGALR, called P03 for short). Spermidine (SPD) was used as a positive control, as the beneficial effects of SPD, including antioxidant and anti-inflammatory activity, have been reported in previous studies [21,22]. Moreover, SPD has been used as a dietary supplement in various preclinical models [23]. Therefore, we carried out this study to investigate the preventive effects of P03 supplementation against binge alcohol exposure and its potential mechanisms.

## 2. Results

### 2.1. P03 Incubation Increases the Viability of L02 Cells Damaged by H_2_O_2_

In Figure 1A, the survival rate of L02 cells gradually decreased as the concentration of H_2_O_2_ continued to increase. The survival rate of L02 cells was approximately 50% when the cells were incubated at the concentration of 600 μM H_2_O_2_ for 4 h. The survival rates of L02 cells in the P03 groups were significantly higher than that of the H_2_O_2_ group, as shown in Figure 1B. The concentrations of 5 and 10 μM of P03 were selected for further study. In Figure 1C–H, it can be seen that the cell cycle of L02 cells in the H_2_O_2_ group was arrested in the G2-M phase, with a percentage of 55.42% in the G1 phase, 10.71% in the S phase and 34.29% in the G2 phase of cells compared to that in the control group. In the cells of the 5 and 10 μM P03 group, the percentage of the G1 phase of cells increased to 62.64% and 69.59%, and the ratios of the G2 phase were 25.05% and 20.11%, respectively. As shown in Figure 1I–Q, the results displayed a considerable increase in the ratio of apoptotic cells damaged by H_2_O_2_ compared to the control group. The total percentage of dead cells after H_2_O_2_ treatment was 22.13%, containing 21.28% apoptotic cells and 0.84% necrotic cells. In contrast, there was higher viability in the cells of the 5 μM P03 and 10 μM P03 groups than that of H_2_O_2_ group, as the proportions of apoptotic cells in the 5 μM P03 and 10 μM P03 groups were reduced to 16.46 and 8.90%.

### 2.2. P03 Supplementation Relieves Oxidative Stress in L02 Cells Damaged by H_2_O_2_

L02 cells were labeled with DCFH-DA (2,7-dichlorodihydrofluorescein diacetate) to detect the levels of intracellular ROS with flow cytometry. The mean intensity of DCF (2,7-dichlorodihydrofluorescein) fluorescence was significantly increased to above eight-fold levels compared to that of the control group when L02 cells were exposed to H_2_O_2_ (Figure 2B,G). The fluorescence intensity in the spermidine and P03 group was significantly lower at one to four-fold than that of the H_2_O_2_ group. The quantification of ROS relative units further confirmed that the ROS levels of the L02 cells in the P03 group decreased compared to the H_2_O_2_ group.

### 2.3. P03 Supplementation Attenuates Acute Ethanol-Induced Hepatomegaly

The experimental process is revealed in Figure 3A. The hepatotoxicity of a xenobiotic agent, such as alcohol, is positively associated with the ratio of wet liver weight to body weight. In Figure 3B, there was no significant difference in the body weight between each group within 5 weeks. Organ coefficient is the ratio of organ weights and body weights. As shown in Figure 3C, the organ coefficients of the livers in the ethanol group were considerably higher than those of the control mice after the intragastrical administration of 50% ethanol. The organ coefficient was 1.20 ± 0.07 in the ethanol group. In contrast, the organ coefficient was reduced to 1.16 ± 0.02 in the spermidine group, 1.10 ± 0.06 in the 10 mg/kg P03 group, and 1.08 ± 0.09 in the 20 mg/kg P03 group, respectively.

### 2.4. P03 Supplementation Significantly Reduces Liver Injury Induced by Ethanol

As demonstrated in Figure 3D–F, increased levels of ALT, AST and γ-GGT were detected in the ethanol group, suggesting the damage to liver parenchyma. Levels of ALT, AST and γ-GGT in the P03 group were significantly lower than those in the ethanol group. As another indicator of liver injury, we also observed the ratios of AST vs. ALT. In the mice of the ethanol group, the ratios of AST vs. ALT were even significantly higher at two-fold over control levels, and decreased levels were observed in the spermidine and P03 group relative to the ethanol group (Figure 3G).

### 2.5. P03 Supplementation Alleviates Oxidative Stress Induced by Binge Alcohol Exposure

As shown in Figure 4A,B, levels of glutathione peroxidase (GPx) and superoxide dismutase (SOD) decreased in the mice of the ethanol group. GPx levels in the spermidine and P03 group were higher than those of the ethanol group, but it was not significant. In mice of the spermidine or P03 group, SOD levels increased significantly compared to the ethanol group.

The results of the immunoblots for nuclear erythroid 2-related factor 2 (Nrf2), heme oxygenase-1 (HO-1) and NADPH quinineoxidoreductase-1 (NQO-1) are demonstrated in Figure 4C–H. Acute alcohol exposure did not affect HO-1 and NQO-1 expression, but tended to promote the nuclear translocation of Nrf2. Compared to the ethanol group, both SPD and 20 mg/kg P03 supplementation could significantly upregulate the protein expression of HO-1. Supplementation with 20 mg/kg P03 significantly increased the nuclear translocation of Nrf2 relative to the ethanol group. The protein expression of NQO-1 could be elevated by SPD and P03 pretreatment relative to the ethanol group; however, the differences were insignificant.

### 2.6. P03 Supplementation Attenuates Alcohol-Induced Lipid Accumulation in Rodent Livers

Liver morphology was assessed in Oil Red O-stained tissue sections. Lipid droplets in frozen liver sections were stained red, while cells and the nucleus were stained blue. As illustrated in Figure 5A–F, the stained tissue in the ethanol group exhibited pronounced macrovesicular steatosis relative to those of the control group. The stained tissue in the spermidine and P03 group presented smaller and fewer lipid droplets than the ethanol group. As presented in Figure 5G,H, the levels of total cholesterol (TC) and tri-glyceride (TG) in the ethanol group were significantly elevated in the ethanol group relative to the control group. Notably, TC and TG levels were reduced in the spermidine and P03 group compared to levels in the ethanol group. In Figure 4I, levels of malondialdehyde (MDA) in the ethanol group increased significantly and were five-fold higher than those relative to the control group. In the spermidine and P03 group, MDA levels were significantly lower than in the ethanol group.

### 2.7. P03 Supplementation Alleviates Hepatitis Induced by Binge Alcohol Exposure

The results of H&E staining are shown in Figure 6A–E. Acute alcohol administration induced evident histological changes in the liver, mainly manifested as microvesicular steatosis and neutrophil infiltrations around the central vein. The SPD or P03 pre-treatment could ameliorate the hepatic inflammatory response. In the 10 and 20 mg/kg P03 group, inflammatory infiltrations and microvesicular steatosis were further attenuated in a dose-dependent manner. As demonstrated in Figure 6F–I, significant increases (~1.5 to 1.7-fold) in the levels of interleukin 6 (IL-6), interleukin 1 beta (IL-1β), tumor necrosis factor alpha (TNF-α) and lipopolysaccharide (LPS) were observed in the ethanol group relative to the control group, indicating that binge alcohol exposure caused hepatic injury and hepatitis. The levels of IL-6, IL-1β, TNF-α and LPS were reduced significantly in the spermidine or P03 group compared to the ethanol group.

The results of the immunoblots for the protein expression of nuclear factor kappa B (NF-κB) in the cytoplasm and the nucleus are illustrated in Figure 7A–D. Compared to the control group, acute alcohol exposure led to a significant increase in NF-κB translocation, which was significantly inhibited in the SPD and 20 mg/kg P03 group. Meanwhile, compared to the ethanol group, the 10 mg/kg P03 supplementation had few effects on NF-κB translocation.

### 2.8. P03 Supplementation Prevents Apoptosis of Hepatocytes Induced by Binge Alcohol Exposure

To investigate cell apoptosis in the liver, B-cell lymphoma-2 (Bcl-2), Bcl-2-associated X (Bax), cytochrome C (Cyto-C), Caspase-3, and poly ADP-ribose polymerase (PARP) expression were immunoblotted. As illustrated in Figure 8A–D, the Bcl-2/Bax ratio in the mice of the ethanol group was reduced two-fold compared to the control group. Compared to the ethanol group, the SPD and 20 mg/kg P03 group exhibited a slight increase in Bcl-2/Bax ratio, but the differences were insignificant. Meanwhile, the protein expression of Cytochrome C, Caspase 3 and PARP in the ethanol group was significantly elevated by 117.8 ± 6.24, 121.8 ± 9.62 and 127 ± 6.97%, respectively, compared to the control group (Figure 8E–G). In the spermidine and 20 mg/kg P03 group, the expression of Cytochrome C, Caspase 3 and PARP was significantly reduced relative to the ethanol group.

### 2.9. The Hepatoprotective Effects of P03 Supplementation May Be Related to The Maintenance of Mitochondrial Function

Mitochondria are highly dynamic organelles undergoing coordinated cycles of fission and fusion, referred to as ‘mitochondrial dynamics’, in order to maintain their shape, selective degradation, and transport. It has been reported that mitochondrial dynamics play an important role in regulating mitochondrial functions, such as apoptosis, calcium handling and energy production [24]. To assess whether P03 has an effect on mitochondrial dynamics, mitochondrial dynamics-related proteins in the liver lysates were immunoblotted, including peroxisome proliferator-activated receptor gamma coactivator-1 alpha (PGC-1α), optic atrophy 1 (OPA1), mitofusin-2 (Mfn-2) and dynamin-related protein 1 (Drp-1). As depicted in Figure 9A–D, oral alcohol administration had no effect on the protein expression of Drp-1, Mfn-2, or OPA1, but led to a significant reduction in PGC-1α expression relative to the control group. Although SPD and P03 pretreatment showed potential to reduce Drp-1 levels and elevate OPA1 expression, there were no significant differences compared to the ethanol group. Mfn-2 protein levels were significantly increased in the SPD and 20 mg/kg P03 group relative to the ethanol group. In the SPD and 20 mg/kg P03 group, the significant elevation of PGC-1α was observed compared to the ethanol group (Figure 9E).

### 2.10. The Hepatoprotective Effects of P03 Supplementation on Autophagy-Related Proteins in Rodent Livers

Autophagy-related protein was immunoblotted to explore whether the hepatoprotective effect of P03 is related to autophagy. First, as illustrated in Figure 10A,B, binge alcohol exposure led to a slight increase in Beclin-1 protein levels, which were significantly elevated in the SPD, 10 mg/kg, and 20 mg/kg P03 groups compared to the ethanol group. Second, as depicted in Figure 10D–E, neither Parkin nor PTEN-induced putative kinase 1 (PINK1) were significantly affected by acute alcohol exposure. Compared to the ethanol group, the significant differences in the protein levels of PINK1 were only observed in the SPD group. Third, the protein levels of sequestosome-1 (p62) exhibited a slight reduction in the ethanol group in comparison to the control group (Figure 10C). Compared to the ethanol group, hepatic p62 levels were significantly reduced in the 20 mg/kg P03 group, whereas few changes were observed in the SPD and 10 mg/kg P03 group. Fourth, microtubule-associated protein light chain 3 (LC3) was involved in the formation of autophagosomes. It is known that LC3-I is hydrolyzed and transformed into LC3-II when autophagy is activated. The LC3-II/LC3-I ratio and the enhancement of LC3-II have been widely used as markers of autophagy. Our present study revealed that heavy alcohol consumption did not affect the hepatic LC3-I protein levels, but significantly elevated the hepatic LC3-II levels (Figure 10F,G). Compared to the ethanol group, 20 mg/kg P03 supplementation also exhibited a potential to increase hepatic LC3-II expression. In Figure 10H, a modest increase in the LC3-II/I ratio was observed in the ethanol group compared to the control group. SPD and 20 mg/kg P03 supplementation have the potential to increase the LC3-II/I ratio; however, there were no significant differences compared to the ethanol group.

## 3. Discussion

Excessive ethanol intake can lead to liver injury, which can be cytologically manifested as hepatocyte swelling, cell membrane rupture, mitochondrial swelling, the leakage of cellular contents, and inflammatory cell aggregation. Excessive ethanol brings more ROS to the liver, which results in oxidative stress, inflammation and cell apoptosis. The results of this study suggest that P03 supplementation effectively alleviates H_2_O_2_-induced cellular damage in the L02 cells, characterized as increases in cell viability ratios, reduced ROS levels, decreases in the ratios of apoptosis, and G2-M phase arrest in the L02 cells damaged by H_2_O_2_.

Hepatocytes swell and necrosis is triggered in response to severe oxidative stress, releasing enzymes such as ALT, AST and γ-GGT into the blood. In this study, we found that serum levels of AST, ALT and γ-GGT were significantly elevated in the ethanol group relative to the control group. Dietary supplementation with P03 could significantly reduce ALT, AST and γ-GGT levels.

In human eukaryotic cells, the metabolism of ethanol leads to the accumulation of ROS, which are produced mainly from the respiratory chain in mitochondria, the cytochrome P4502E1 in the endoplasmic reticulum, and Kupffer cells [25]. Ethanol exposure leads to LPS stimulation [26]. Serum levels of LPS are associated with the severity of hepatic injury [27,28,29]. Additionally, it has been reported that both TNF-α and IL-6 are substantially elevated in the circulation of patients with acute alcoholic hepatitis [30]. IL-1β is also an inflammatory cytokine, and its levels are significantly increased in the liver and the serum in patients with alcoholic hepatitis [31]. As reported by the studies mentioned above, our results also found that acute alcohol intake significantly augmented the levels of IL-6, IL-1β, TNF-α, and LPS in the serum. SPD and P03 supplementation significantly inhibited the elevation of the indicated cytokines and LPS. NF-κB plays an important role in regulating the expression of several inflammatory factors [32], and alcohol consumption stimulates NF-κB-induced inflammation [33]. Our results showed that compared to the control group, acute alcohol intake significantly elevated the NF-κB translocation, confirming the activation of NF-κB signaling. In contrast, SPD and P03 supplementation exhibited NF-κB translocation in a dose-dependent manner, but the differences were observed to be significant only in the SPD and 20 mg/kg P03 group in comparison to the ethanol group. Histopathological analysis showed that SPD and P03 were effective in ameliorating hepatic inflammatory cell infiltration around the central vein. These results indicate that the inflammatory response caused by excessive ethanol intake could be suppressed by SPD and P03 supplementation.

Numerous investigations have revealed that a considerable increase in blood triglyceride and total cholesterol levels is associated with hepatic steatosis [34,35]. Hepatic steatosis is characterized by excessive triglyceride accumulation in the cytoplasm of hepatocytes. There is evidence that heavy alcohol consumption can lead to hepatic lipid accumulation, which is consistent with our results [36]. In this study, we observed significant reduced levels of TG and TC in the SPD and P03 group when compared to the ethanol group. Because hepatic steatosis is associated with lipid peroxidation in mice [37], we tested the malondialdehyde (MDA) levels in the liver. MDA is the natural product of lipid peroxidation. Our results showed that hepatic MDA levels were significantly increased in the ethanol group compared to the control group. In contrast, SPD and P03 supplementation significantly reduced the hepatic MDA levels compared to the ethanol group. The histopathological analysis further supported that P03 supplementation could prevent hepatic steatosis in mice with acute alcohol exposure.

Nrf2 plays a crucial role in the defense against oxidative stress. It maintains intracellular redox balance and protein homeostasis by regulating the expression of downstream antioxidant enzymes and exerts biological functions such as antioxidants [38]. Nrf2 is physiologically coupled to keap1 and ubiquitously expressed in various tissue and cell types. When oxidative stress occurs, Nrf2 translocates into the nucleus. It binds to the antioxidant response element (ARE) in the nucleus to initiate the transcription of downstream antioxidant proteins, which increases the expression of antioxidant proteins [39]. HO-1 exerts an antioxidant effect by catalyzing heme degradation [40]. NQO1 can catalyze a reduction in quinones and their derivatives, preventing further redox reactions [41]. Accordingly, promoting the nuclear translocation of Nrf2 may be a possible way to reduce oxidative stress for acute alcoholic liver injury. Our results showed that heavy alcohol intake led to a slight increase in Nrf2 translocation compared to the control group. However, neither HO-1 nor NQO-1 were affected in the ethanol group, likely due to its depletion to defend against oxidative stress. In contrast, P03 supplementation significantly increased hepatic HO-1 protein levels and Nrf2 translocation compared to the ethanol group. Meanwhile, hepatic SOD activity was significantly higher in the SPD and P03 group than in the ethanol group. For GPx activity, statistical differences were observed only in the 20 mg/kg P03 group compared to the ethanol group. The current results show that the antioxidant defense could be significantly activated by 20 mg/kg P03 supplementation.

A healthy mitochondrion is key to energy balance in the liver [42]. Active mitochondria can continuously change their shapes through fission and fusion, and some studies clearly demonstrate that the imbalance of this process has a detrimental impact on mitochondrial function [24]. Mitochondrial fusion is mainly mediated by the proteins of Mfn-2 and OPA1, while mitochondrial fission and biogenesis are mediated by the proteins of Drp-1 and PGC-1α [43,44]. It has been reported that several physiological indicators of mitochondrial dysfunction are associated with the imbalance of mitochondrial dynamics, such as a decrease in mitochondrial membrane potential and increased ROS formation [45]. Furthermore, there has been evidence in previous studies of the important role of ROS in the regulation of mitochondrial dynamics, suggesting a link between cellular redox homeostasis and the regulation of mitochondrial dynamics [46]. ROS at physiological levels can act as a signaling molecules to regulate mitochondrial dynamics on mitochondrial fusion and fission proteins. When a large amount of ROS are released and the antioxidant system is not activated to scavenge excessive ROS in the cells, this promotes mitochondrial fragmentation, swelling or shortening. Intracellular antioxidants such as oxidized glutathione (GSSG) are able to enhance the antioxidant response through Mfn-2-dependent mitochondrial fusion [47]. Moreover, Mfn-2 knockdown elevates ROS levels [48]. PGC-1α not only induces mitochondrial biogenesis, but is also involved in regulating cellular antioxidant defense. The upregulation of PGC-1α has been found to be essential to prevent cells from mitochondrial dysfunction and oxidative damage [49,50]. The present results showed that heavy alcohol intake did not affect hepatic Drp-1, OPA1 or Mfn-2 protein levels, but caused a significant decrease in hepatic PGC-1α levels. Meanwhile, SPD and 20 mg/kg supplementation significantly elevated hepatic Mfn-2 and PGC-1α levels. These findings suggest that the antioxidant effects of P03 supplementation may be related to the enhancement of mitochondrial dynamics.

Excessive ROS generated from ethanol metabolism could damage mitochondria, causing the dysfunction or rupture of mitochondria, releasing DAMPs (damage-associated molecular patterns) such as mtDNA which aggravate the inflammatory response [51]. The damaged mitochondria which cannot be repaired release mitochondrial proteins into the cytoplasm, including cytochrome C, which promotes Caspase-3 activation and chromatin degradation [52]. Bcl-2 family proteins play important regulatory roles in the mitochondrion-mediated endogenous apoptosis [53]. However, our results found that there are no significant differences in the Bcl-2/Bax ratio between the control group and the ethanol group, although a decreased Bcl-2/Bax ratio was observed in the ethanol group. Therefore, we immunoblotted the downstream proteins including cyto C, caspase-3 and PARP. Our results show that the protein levels of cyto C, caspase-3, and PARP were significantly increased in the ethanol group compared to the control group. In contrast, this increase was significantly reversed via the supplementation of SPD and 20 mg/kg P03. These results suggest that mitochondrion-mediated apoptosis was involved in acute alcoholic liver injury, and 20 mg/kg P03 supplementation could promote the anti-apoptotic ability of hepatocytes.

As high levels of ROS damage biological macromolecules, autophagy has an essential role in the removal of abnormal or damaged liver proteins that would otherwise accumulate and lead to hepatotoxicity. Previous investigations have shown that autophagy is activated in response to oxidative stress to protect cells from apoptosis [54]. Mitochondrial autophagy (mitophagy) is a particular form of autophagy that selectively degrades damaged mitochondria through autophagy. The PINK1/Parkin pathway is typical of chaperone-mediated autophagy. Autophagy mediated by the PINK1/Parkin pathway in eukaryotic cells can be recapitulated as follows [55,56,57,58,59,60]. PINK1 accumulates on the outer membrane of the damaged mitochondria, recruiting Parkin to the same location for autophagy initiation. Parkin then ubiquitinates substrates, and the ubiquitinated substrates binds to the autophagy receptor protein p62. Then, the receptor protein p62 binds to LC3 to form the autophagy vesicles. Finally, autophagy vesicles are combined with the lysosome and degraded by hydrolase in the lysosome. Mitochondria are one of the organelles most susceptible to ethanol exposure [42]. Given that autophagy is activated in the liver after oxidative injury to remove damaged proteins and excess ROS [61], we preliminarily explored the effects of P03 supplementation on the PINK1/Parkin pathway. First, Beclin-1 is an essential autophagy protein that functions in autophagosome formation [62]. In addition, it was found that the expression of antioxidant enzymes, including Cu-ZnSOD and catalase, was significantly reduced in response to the inhibition of autophagy caused by p62 accumulation, resulting in the impairment of antioxidant defense [63]. Our results show that hepatic Beclin-1 protein levels were significantly elevated by SPD and P03 supplementation compared to the ethanol group, whereas significant differences in hepatic p62 protein levels were only observed in the 20 mg/kg P03 group in comparison to the ethanol group. Second, we found that P03 supplementation had no effect on Parkin and PINK1 protein expression. Third, enhanced LC3-II expression and elevated LC3-II/LC3-I ratios were observed in the 20 mg/kg P03 group compared to the ethanol group but without statistical significance. The results only explained that autophagy is involved in the mechanism of the hepatoprotective effects of P03 supplementation. More systematic verifications should be performed in the future.

In our previous study, in vitro experiments revealed that a major fraction of the peptide was taken up by cells via endocytosis and was entrapped in lysosomes [20], but further studies are needed to determine the pharmacokinetic characteristics of the peptide in vivo. Despite their limitations, our results suggest that dietary supplementation with P03 can considerably ameliorate acute alcoholic liver injury in mice. Its mechanisms may be relevant to antioxidant defense and mitochondria protection. Spermidine was more effective than P03 in this study. However, one study revealed that polyamines such as spermidine may favor tumor growth yet inhibit colon carcinogenesis once cancer has developed [64]. Additionally, polyamine levels have previously been associated with procarcinogenic activities [23]. It has been reported that elevated levels of acetylated polyamine can serve as biomarkers for different tumor types [65].

In conclusion, our study suggests that a sea cucumber-derived peptide could be one of the potential candidates from food for the amelioration of acute alcoholic liver injury. More safe and active antioxidants from natural foods can serve as dietary supplements and should be explored for liver protection against severe hepatic injury caused by heavy alcohol consumption.

## 4. Materials and Methods

### 4.1. Materials and Reagents

Spermidine was purchased from Sigma-Aldrich (St. Louis, MO, USA). Ethanol was obtained from Zhiyuan Chemical Reagent Co. Ltd. (Tianjin, China). Peptide was isolated and identified as previously described [66]. Briefly, peptide was isolated by liquid chromatography tandem mass spectrometry using a reverse-phase nanocolumn (RP-nano-LC-MS/MS). Peptide sequences were identified and searched with ProteinPilot Software (AB SCIEX, Framingham, MA, USA) against a translated protein database of A. japonicus nucleotide sequences from the National Center for Biotechnology Information (NCBI). P03 was synthesized with solid-phase synthesis with a purity of >95% from Shanghai Top-peptide Biotechnology Co., Ltd. (Shanghai, China). The certificate of analysis and MS spectrum for P03 were showed in Appendix A. The mitochondrial membrane potential assay kit with JC-1(#C2006), the cell cycle and apoptosis analysis kit (#C1052), the annexin V-FITC apoptosis detection kit (#C1062L), and the reactive oxygen species assay kit (#C0033) were purchased from Beyotime Biotechnology (Shanghai, China).

### 4.2. Cell Viability Assay

Hepatic cell line L02 (human liver cells) was obtained from the Chinese Academy of Science (Shanghai, China). L02 cells were cultured with Dulbecco’s Modified Eagle’s Medium (DMEM) supplemented with 10% fetal bovine serum (Waltham, MA, USA) and 1% penicillin/streptomycin solution (Hyclone, Logan, UT, USA) and were incubated at 37 °C in a humidified atmosphere incubator (Thermo Scientific, Rockford, IL, USA) with 5% CO_2_. L02 cells were seeded in 96-well plates at a density of 8000 cells in 100 µL medium per well and incubated for 24 h. Cells were then incubated with 0, 5, 10, 50, 100, 200 μM of P03 in DMEM supplemented with 10% FBS for 24 h. Subsequently, cells were incubated in fresh DMEM supplemented with 10% FBS with or without 600 μM H_2_O_2_. Four hours later, 5 μL of MTT (5 mg/mL) solution was added in 100 μL DMEM without 10% FBS and incubated at 37 °C for 4 h. After removing the supernatant, 100 μL of dimethyl sulfoxide was added to dissolve the formazan crystals. Absorbance was measured at a 570 nm wavelength with a microplate reader (Bio Tek Instruments, Winooski, VT, USA).

### 4.3. Flow Cytometric Analysis

L02 cells were cultured in 6-well plates at a density of 2 × 10^5^ cells in 2000 µL medium per well for 24 h. Cells were then incubated with 5, 10 μM of P03 or 10 nM spermidine for 24 h after being incubated for 4 h with 600 μM H_2_O_2_. The mitochondrial membrane potential assay kit, cell cycle and apoptosis analysis kit, annexin V-FITC apoptosis detection kit and reactive oxygen species assay kit were performed with flow cytometry (CytoFlex, Beckman, Brea, CA, USA) according to the manufacturer’s instruction.

### 4.4. Animal Experiments

Animal experimental procedures and animal welfare were approved by the Animal Research Ethics Committee of Guangdong Pharmaceutical University in China (approval no. gdpulacspf2017413-1). Six-week-old C57BL/6 male mice were purchased from Guangdong Medical Laboratory Animal Center. The specific pathogen-free mice were housed in polycarbonate metabolic cages in a temperature-controlled room at 25 °C with 12 h day–night rhythm. All mice were fed on a commercial diet for 5 weeks and had free access to food and water. After a week of acclimatization, 25 mice were randomly separated into five groups of five mice each: (1) control; (2) ethanol; (3) 1 mg/kg b.w. spermidine; (4) 10 mg/kg b.w. P03; (5) 20 mg/kg b.w. P03. Spermidine and P03 were administered orally every other day over 35 days (Figure 3A). On day 36, all mice except those in the control group received a single dose of 12 mL/kg b.w. of 50% *v/v* ethanol by intragastric administration, whereas those in the control group were administered with an equal amount of saline. Six hours later, the animals were anesthetized with ether for 1 h, and blood samples were taken through intracardiac puncture. Then, mice were euthanized via cervical dislocation. A portion of each liver tissue was removed from the same location and placed in 4% paraformaldehyde solution, and the rest of the tissue was frozen at −80 °C for subsequent analysis.

### 4.5. Measurement of Liver Coefficient

The liver coefficient was determined by the ratio of liver wet weight to body weight (g/g), respectively. Values are expressed as liver coefficient = liver weight (g)/body weight (g) × 100%.

### 4.6. Enzyme-Linked Immunosorbent Assay of ALT, AST, γ-GGT, TG, TC, LPS and Inflammatory Cytokines

Blood samples were obtained from the heart of mice after ethanol gavage, and over 100 μL of serum samples were separated by centrifuge and stored at −80 °C for analysis. The levels of alanine aminotransferase (ALT), aspartate aminotransferase (AST), γ-glutamyl transpeptidase (γ-GGT), triglyceride (TG), total cholesterol (TC), lipopolysaccharide (LPS), and IL-6, IL-1β and TNF-α were determined with ELISA kits (Meimian Biotechnology, Yancheng, Jiangsu, China) according to the manufacturer’s instructions.

### 4.7. Liver Histopathology

Liver tissues were fixed overnight in 4% paraformaldehyde, embedded in paraffin, and sectioned at a thickness of 5 μM. The sections were stained with hematoxylin and eosin. Sirius red staining was performed with the kits obtained from Baso (Zhuhai, China). Lipid droplets in frozen liver sections were stained with the Oil Red O Staining Kit (Servicebio, Wuhan, China). All sections were visualized with Bioquant Osteo.

### 4.8. Total, Cytoplasmic, and Nuclear Protein Extraction

Total proteins were extracted from liver tissue in lysis buffer. The supernatant was collected after 12,000× *g* centrifugation at 4 °C and stored at −20 °C for analysis. Nuclear and cytoplasmic proteins were fractionated with the kits from Beyotime Biotechnology (Shanghai, China).

### 4.9. Antioxidant Enzymes and MDA Quantification

For the determination of the antioxidant enzymes, proteins of liver tissue were prepared as indicated. The activities of glutathione peroxidase (GPx) and superoxide dismutase (SOD) were detected independently with the total glutathione peroxidase assay kit and the total superoxide dismutase assay kit (both from Beyotime, Shanghai, China). The levels of malondialdehyde (MDA) were detected with the lipid peroxidation MDA assay kit (Beyotime, Shanghai, China). All kits were performed according to the manufacturer’s instructions.

### 4.10. Immunoblotting

Total proteins were isolated from liver tissue as described previously. Specific antibodies against Nrf2, HO-1, NQO-1, Bcl-2, Bax, Cyto C, PARP, Caspase-3, p62, Beclin-1, Parkin, LC3, PGC-1α, OPA-1, Mfn-2, Drp-1, NF-κB, β-actin and Lamin B1 were purchased from Cell Signaling Technology (Danvers, MA, USA). Specific antibodies against PINK1 were purchased from Abcam (Cambridge, MA, USA). β-actin, and Lamin B1 were used to quantify protein expression. Densitometric quantifications of indicated proteins were analyzed by Image J (NIH, Bethesda, MD, USA). Quantifications of liver sections were analyzed by Image Pro plus (Media Cybernetics, Rockville, MD, USA).

### 4.11. Statistical Analysis

GraphPad Prism (Version 7.0) for Microsoft Windows (GraphPad Software, San Diego, CA, USA) was used for statistical analysis and graph generation. Comparisons were made with one-way ANOVA with Bonferroni post hoc correction to assess the differences among multiple groups, respectively. All experiments were performed at least three times. *p* values < 0.05 were considered to be statistically significant.

## Figures and Tables

**Figure 1 molecules-27-05839-f001:**
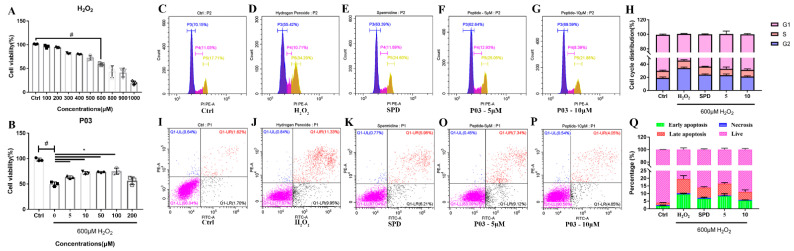
In vitro assays. (**A**) Viability of L02 cells with different concentrations of H_2_O_2_. (**B**) Viability of L02 cells damaged by H_2_O_2_ after 24 h incubation of different concentration of peptide. (**C**–**G**) The cell cycle distribution in L02 cells with flow cytometry. (**H**) Quantitative analysis of cell cycle distribution in each group. (**I**–**P**) The apoptosis ratios of L02 cells in different groups with flow cytometry. (**Q**) Quantitative analysis of the percentage of cell apoptosis in each group. Comparisons were made with one-way ANOVA. Values were expressed as the percentage of the control and represented as mean ± SD. (*n* = 3) (compared to Ctrl, ^#^ *p* < 0.05; compared to 0, * *p* < 0.05) (*n* = 3).

**Figure 2 molecules-27-05839-f002:**
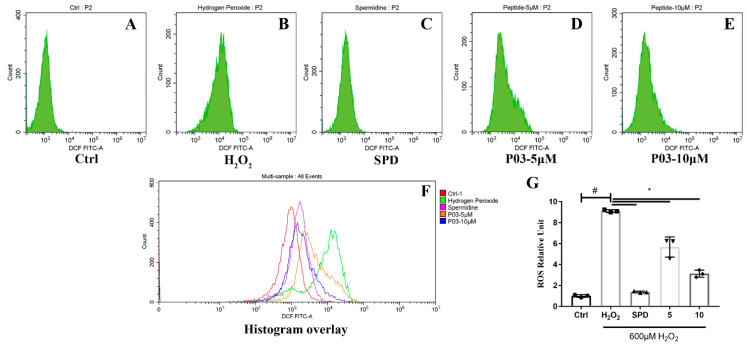
The ROS levels of L02 cells. (**A**–**E**) The mean DCF fluorescence intensity of L02 cells in different groups with flow cytometry. (**F**) Histogram overlay of each group. (**G**) Quantitative analysis of ROS relative units in each group. Comparisons were made with one-way ANOVA. Values were expressed as the ratio of the control and represented as mean ± SD (*n* = 3) (compared to Ctrl, ^#^
*p* < 0.05; compared to H_2_O_2_, * *p* < 0.05) (*n* = 3).

**Figure 3 molecules-27-05839-f003:**
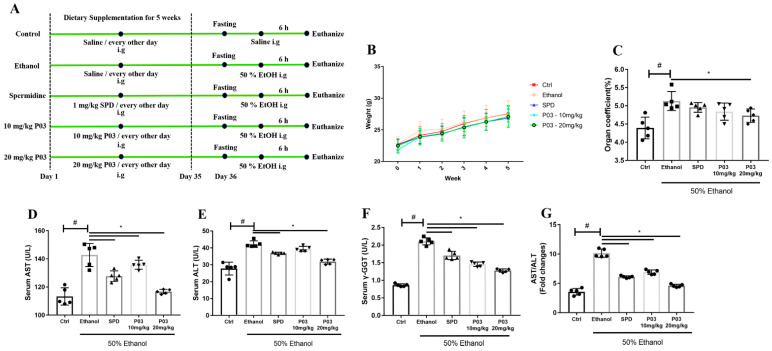
Peptide P03 attenuated hepatic injury induced by binge alcohol exposure. (**A**) Schematic diagram depicting the experimental approach to evaluate the effect of P03 supplementation on mice with binge alcohol exposure. (**B**) Body weight change. The data in week 0 represent the body weight of mice during acclimation. (**C**) Organ coefficients of acute ethanol-intoxicated mice in different groups. The organ coefficient  =  organ weight/body weight. (**D**–**G**) Serum alanine aminotransferase (ALT), aspartate aminotransferase (AST), gamma-glutamyl transpeptidase (γ-GGT) levels and ratios of AST vs. ALT in different groups. Values are expressed as mean ± SD. (Compared to Ctrl, ^#^
*p* < 0.05. Compared to ethanol, * *p* < 0.05) (*n* = 5).

**Figure 4 molecules-27-05839-f004:**
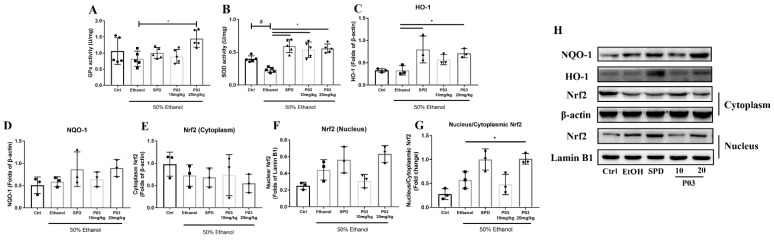
Alterations in antioxidant enzymes in rodent livers and immunoblots of HO-1, NQO-1 and Nrf2. (**A**,**B**) Quantification of GPx and SOD in different groups (*n* = 5). (**C**,**D**) Hepatic levels of NQO-1 and HO-1 protein. (**E**–**G**) Relative protein levels of Nrf2 in nucleus and cytoplasm in each group (*n* = 3). Densitometric quantification of HO-1, NQO-1, cytoplasmic and nuclear Nrf2 were performed and normalized to β-actin levels (for the cytosolic factions) or Lamin B1 (for the nuclear fractions). (**H**) Western blot analysis of indicated proteins in liver lysates of different groups. Values are represented as mean ± SD and plotted as fold-change. (Compared to Ctrl, ^#^
*p* < 0.05. Compared to ethanol, * *p* < 0.05).

**Figure 5 molecules-27-05839-f005:**
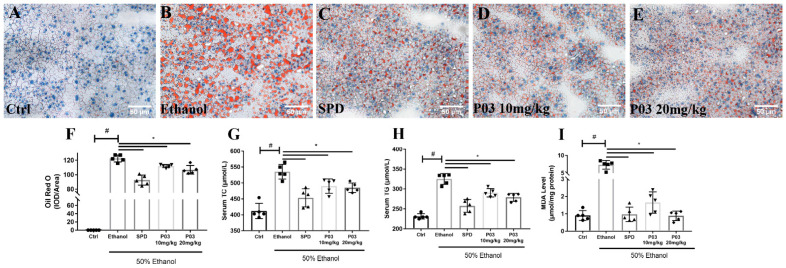
Alterations in lipid metabolism in mice with binge alcohol exposure. (**A**–**E**) Representative images of histological liver sections after Oil Red O staining. Sections were visualized using Bioquant Osteo (20×) and quantified using Image Pro plus software. (**F**) The corresponding quantification of Oil Red O staining in different groups. (**G**,**H**) Serum triglycerides and total cholesterol levels in different groups. (**I**) The levels of hepatic malondialdehyde (MDA) were detected colorimetrically in different groups. Comparisons were made with one-way ANOVA. Values are represented as mean ± SD. (Compared to Ctrl, ^#^
*p* < 0.05. Compared to ethanol, * *p* < 0.05) (*n* = 5).

**Figure 6 molecules-27-05839-f006:**
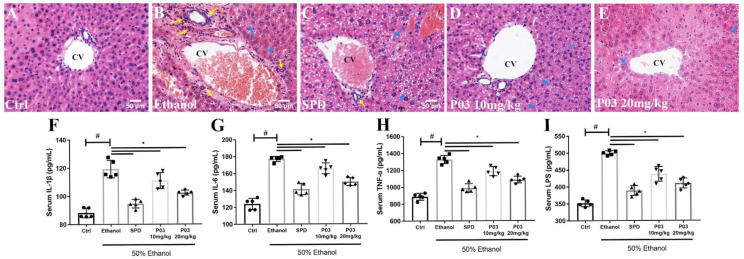
Anti-inflammatory effects of P03 in mice with binge alcohol exposure. (**A**–**E**) Representative images of histological liver sections after H&E staining. Sections were visualized using Bioquant Osteo (20×). Yellow arrows represent neutrophil infiltration surrounding the central vein (CV); blue arrows represent microvesicular steatosis. Scale bar = 50 μM. (**F**–**I**) Serum levels of inflammatory cytokines and LPS in different groups. Comparisons were made with one-way ANOVA. Values are represented as mean ± SD. (Compared to Ctrl, ^#^
*p* < 0.05. Compared to ethanol, * *p* < 0.05) (*n* = 5).

**Figure 7 molecules-27-05839-f007:**
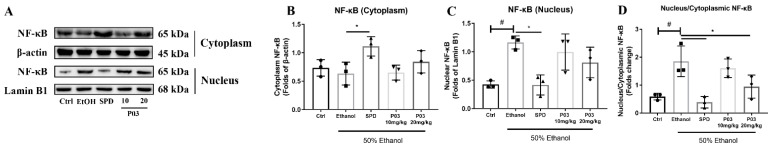
Immunoblots of nuclear and cytoplasmic NF-κB. (**A**) Western blot analysis of NF-κB in different groups. (**B**–**D**) Relative protein levels of NF-κB in nucleus and cytoplasm in each group. Densitometric quantification of nuclear and cytoplasmic NF-κB were performed and normalized to β-actin levels (for the cytosolic factions) or Lamin B1 (for the nuclear fractions). The values are represented as mean ± SD and plotted as fold-change. (Compared to Ctrl, ^#^
*p* < 0.05. Compared to ethanol, * *p* < 0.05).

**Figure 8 molecules-27-05839-f008:**
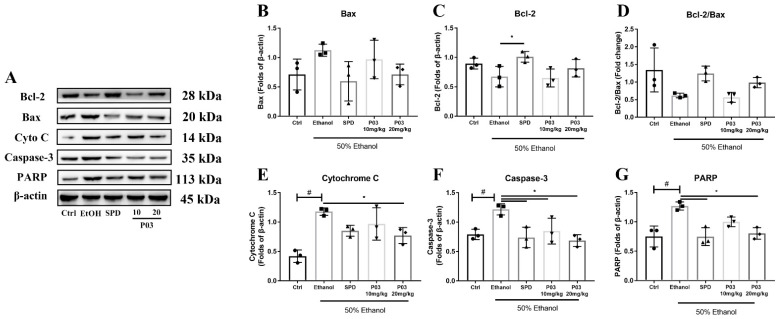
Immunoblots of Bcl-2, Bax, Cytochrome C, Caspase-3 and PARP. (**A**) Immunoblots of indicated proteins. (**B**–**F**) Densitometric quantification of Bcl-2, Bax, Cytochrome C, Caspase-3 and PARP were performed and normalized to β-actin levels. (**G**) The ratios of Bcl-2 vs. Bax were calculated from densitometric analysis. The values represent mean ± SD and are plotted as fold-change. (Compared to Ctrl, ^#^
*p* < 0.05. Compared to ethanol, * *p* < 0.05) (*n* = 3).

**Figure 9 molecules-27-05839-f009:**
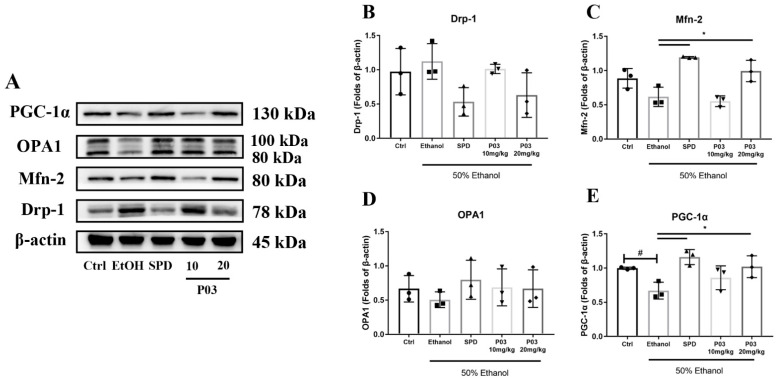
Immunoblots of PGC-1α, OPA-1, Mfn-2, and Drp-1. (**A**) Immunoblots of indicated proteins. (**B**–**E**) Densitometric quantification of PGC-1α, OPA-1, Mfn-2, and Drp-1 were performed and normalized to β-actin levels. The values are represented as mean ± SD and plotted as fold-change. (Compared to Ctrl, ^#^
*p* < 0.05. Compared to ethanol, * *p* < 0.05) (*n* = 3).

**Figure 10 molecules-27-05839-f010:**
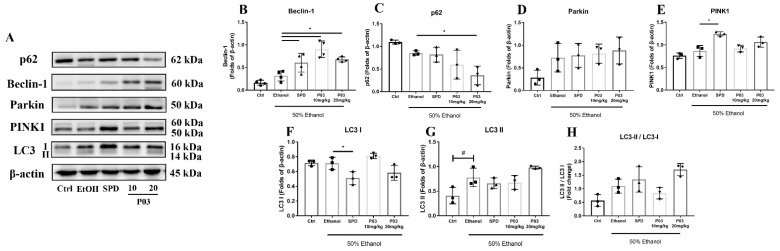
Immunoblots of mitophagy-related proteins. (**A**) Immunoblots of indicated proteins. (**B**–**H**) Densitometric quantification of p62, Beclin-1, Parkin, PINK1 and LC3 (I, II) were performed and normalized to β-actin levels. The values are represented as triplicate experiments ± SD and plotted as fold-change. (Compared to Ctrl, ^#^
*p* < 0.05. Compared to ethanol, * *p* < 0.05) (*n* = 3).

## Data Availability

Data are available from the corresponding author upon request.

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
