# Peer review of "A Peptide HEPFYGNEGALR from Apostichopus japonicus Alleviates Acute Alcoholic Liver Injury by Enhancing Antioxidant Response in Male C57BL/6J Mice"

_molecules, 2022, doi:10.3390/molecules27185839_

Round 1

Reviewer 1 Report

The authors investigated the effect of peptide (P03) from Apostichopus japonicas against acute alcoholic liver injury L02 hepatocytes and mice. Results showed P03 treatment reduced cellular apoptosis and ROS levels induced by H2O2 in vitro. P03 attenuated ethanol-induced hepatic injury as shown by a reduction in liver size and AST, ALT, gamma GGT levels. P03 also reduced alcohol-induced hepatic oxidative stress, lipid accumulation, and inflammation. Authors suggest that the mechanism of P03 might be associated with the reduced apoptotic factor activity, restored mitochondrial dynamics and enhanced mitophagy. Authors conclude that a dietary supplementation of the peptide potentially alleviate acute alcoholic liver injury. Below is my comment.  

 1.      The title should start with “ A peptide HEPFYGNEGALR from…..” to make clear what is your novel effective compound.

2.      In the Abstract (line 30, p1), authors wrote that “…alleviate mitochondrial dysfunction induced by ethanol”; however, authors did not show any direct evidence showing mitochondrial function (ATP synthesis during cellular respiration, an important mitochondrial function). Please mention on this in the discussion if you have no data.

3.      In the Introduction (line 78, p2), authors mentioned that three peptides isolated and identified as anti-oxidant effect in neuroblastoma cells. What did you choose P03 among three peptides in this study?  

4.      In Figure 1 and 2, authors investigated the effect of P03 in H2O2-treated cells. Why did you treat cells with ethanol directly to see the effect against ethanol-induced cellular injury?

5.      Spermidine is a polyamine compound, being heavily positive-charged and having various metabolic functions; please explain why you have a spermidine treatment group in the study.

6.      In Fig.4C-F, Fig. 7B-C, Fig. 8B-G, Fig. 9B-E, Fig. 10B-F, the statistical significance was not found or properly shown in the bar graphs although the corresponding text described there were significant changes by the treatments. Please include individual data as dots in the bar graphs to display high variable data, and perform the statistical analysis again.

7.      In Fig. 7B-C, NF-kB nuclear expression showed too much variation and no significant changes (error bars are overlapped with the comparison group). Include also an N/C ratio of NF-kB expression.

8.      In Fig. 9D, there is no difference in OPA1 expression in all groups-fix the description in the results. Similarly, in Fig. 10, the Parkin, Pink, p62, LC3 levels are not significantly different among treatment groups compared to the ethanol group; however, you described in the results as there is the effect by each treatment, which is scientifically wrong!

9.      In the discussion, authors should discuss on pharmacokinetic characteristics- being administered, how is the peptide absorbed and distributed mostly, and metabolized for its action in the body? Then, how does the peptide interact with particular receptors or enzymes to modify their biological functions??

Author Response

Dear Editors and Reviewers,

  We gratefully appreciate the editors and all reviewers to revise our manuscript entitled “ Peptide from Apostichopus japonicus Alleviates Acute Alcoholic Liver Injury by Enhancing Antioxidant Response and Mitophagy” (Manuscript ID: molecules-1834883). We highly appreciate the reviewer’s and editor’s constructive comments and suggestions on our manuscript, which is really helpful for us.

  We have studied the comments carefully and have accordingly made revisions marked up using the “Track Changes” function in the revised manuscript, which we would like to submit for your kind consideration. As attached, the summary of corrections and the responses to the reviewer’s comments are listed in the Revision Report. Please see the attachment entitled "Response letter".

  We hope that the revision and correction will meet with approval.

Thank you and best regards.

Yours sincerely,

Dr. Hongliang Huang.

Tel: 86 + 13570908699

Address: School of Biosciences & Biopharmaceutics, Guangdong Pharmaceutical University, Guangzhou 510006, China.

Reviewer 2 Report

This is an interesting work where the authors explore the beneficial effects of the treatment with a peptide derived/isolated from Apostichopus japonicus on acute alcohol-induced liver injury. To do so, the authors test their peptide first in vitro by using human liver cell line (L02 cells) to analyze cell viability. Then, the authors explore the effects in a murine model of acute alcohol-induced liver injury. The authors study several cellular pathways: inflammation, oxidative stress and antioxidant defense, apoptosis, mitochondrial function, and mitophagy. This is a wide (and ambitious) range of cellular pathways to explore. Manuscript redaction is not neat or informative (abbreviations appearing before being defined, confusing language, poor figure legends and titles, etc.). The main concern is found in the “Results”. The authors have found some interesting results. However, the authors present some trends found are proposed as statistical significant increases or reductions to propose the conclusions (which is not accurate). Based on this, the authors should tone down the conclusions based on the results found.

There some particular points specifically related to the experimental design that this reviewer wants to point out:

1. The authors should revise the English language, especially how the sentences are a constructed.

2. It is not sufficiently described how the authors isolate this peptide and its sequence. In lines 80-81, it is written: “The sequence of one peptide from three peptides was His-Glu-Pro-Phe-Tyr-Gly-80 Asn-Glu-Gly-Ala-Leu-Arg (HEPFYGNEGALR, called P03 for short)”. What do the authors mean with “on pepetide from three peptides”?

3.  In in vitro viability assays, why did the authors select 5 and 10 microM concentrations? (lines 90-91) In figure 1A, 5 microM had no significantly effect versus control group.

4. This reviewer has not found the reason why Spermidine is used along the manuscript in parallel with the tested peptide.

5. Lines 97-100. What was the percentage of necrotic cells after treatment with P03?

6. Line 112. Please use specify which figure are referring to. For instance, in this case the authors should use “Figure 2B and 2G” instead of Figure 2.

7.  This reviewer strongly recommends including individual values into the bar-plots, because it helps to visualize the distribution of the individual experiments. This may apply to Figures: 2G, 4CDEF, 7BC, 8BCDEFG, 9BCDE, 10BCDEF.

8. Experimental procedure described in Figure 3A and in “Methods” section is confusing. Please clarify. How many animals per group did the authors use? How did the authors decide the peptide dose administered to the mice? And the duration of the treatment?Were there exitus during the treatment? Did the treatment affect any metabolic parameter versus baseline?

9. The designed protocol represents a preventive therapeutic strategy against the acute hepatic injury induced by alcohol. Would the treatment would effective received after the induction of injury?

10. Figure 3B, Y-axis units. Please modify: the units should appear between parenthesis Weight (g)

11. Please define “Organ coefficient” the first time that appears in the text (line 129). Why do the authors use brain? Should it be better to use liver/body weight ratio?

12. Figure titles and legends should be revised and rewritten. For instance: Figure 3, figure 6

13. Please define abbreviations the first time that are found in the manuscript (i.e. ALT, AST, GGT).

14. Figure legend – Figure 3. As far as this reviewer understands, when the authors use the symbol # reference group for the comparison is CTRL, and (*) is used the reference group is “ETHANOL”. Please revise ALL the figure legends, there is an error which is repetitively found.

15. Figure legend – Figure 3. The following belongs to “Methods”: “Figure 3A was created with 137 BioRender.com.

16. Line 151. Which enzymes are the authors referring to? Please the authors should facilitate the reading process.

17. Line 155. When the authors write “increased significantly”, it is redundant to include “*P<0.05”. This has been found along the manuscript. Please revise it.

18. Lines 158-163 – Oxidative stress results. Although increases are observed, there are no significant differences (as shown by the absence of symbols in the figures).

19. How many experiments did the authors performed (n=?)? This reviewer strongly encourages the authors to include the individual data in the bar plots, as done in figures 4AB, and specify in figure legends.

20. Does the lipid accumulation come from de novo lipogenesis?

21. Lines 175-176, Figures 5ABCDE. Would it be possible to quantify Oil Red O staining?

22. How did the authors asses the histological changes? This is poorly explained and demonstrated (microphotographs have no sufficient resolution and detail).Do the authors know whether a score of pathohistological changes exist?

23. Lines 198-199. Is it “expression” or “secretion” of cytokines and LPS?

24. Lines 206-207. The authors state: “In contrast, the decreases were observed in the spermidine or P03 group relative to the ethanol group”. However, differences only were significant for SPD group.

25. Line 240. Proteins of Mitochondrial dynamics appear for the first time in the manuscript. Please define them.

26. Mitophagy – Results. The authors drive to conclusions, which are not coincident with the results shown.

27. Along the Discussion section, again the authors come to conclusions that no coincide with the observed results.

Lines 304-307: not significant differences in Nrf2 translocation were observed in Results section

Lines 315-317: Only differences at the highest dose and on 2 out of 4 of the studied proteins were observed for mitochondrial dynamics

Lines 323-324: Not convincing results, the translocation was not significant.

Lines 329-331: Again for apoptosis

Lines 351-353: and for mitophagy

28. Methods: Why do the authors only study protein expression? What does it happen with gene expression?

29. Methods: Statistical analysis is poorly described. Should it be more suitable a one-way ANOVA with Bonferroni post-hoc correction?

Author Response

(The authors gave the same response as above.)

Round 2

Reviewer 1 Report

Authors responded appropriately to the comments of the reviewers. The authors recommend including a discussion of how the antioxidant effects of P03 are related to the enhancement of mitochondrial dynamics (fission and fusion) and autophagy activity in ethanol-treated hypatocytes. Instead of showing an extensive list of cellular pathways under study, it is important to present an effective therapeutic point of action of a drug (e.g. P03 in this study) on the diseased condition. 

Author Response

Dear Editors and Reviewers,

  We gratefully appreciate the editors and all reviewers to revise our manuscript entitled “ Peptide from Apostichopus japonicus Alleviates Acute Alcoholic Liver Injury by Enhancing Antioxidant Response and Mitophagy” (Manuscript ID: molecules-1834883). We highly appreciate the reviewer’s and editor’s constructive comments and suggestions on our manuscript, which is really helpful for us.

  We have studied the comments carefully and have accordingly made revisions marked up using the “Track Changes” function in the revised manuscript, which we would like to submit for your kind consideration. As attached, the summary of corrections and the responses to the reviewer’s comments are listed in the Revision Report. Please see the attachment entitled “Response letter.”

  We hope that the revision and correction will meet with approval.

Thank you and best regards.

Yours sincerely,

Dr. Hongliang Huang.

Tel: 86 + 13570908699

Address: School of Biosciences & Biopharmaceutics, Guangdong Pharmaceutical University, Guangzhou 510006, China.

Reviewer 2 Report

The authors have addressed the raised comments suggested by this reviewer. 

Author Response

(The authors gave the same response as above.)
